# DetectLLM: Leveraging Log-Rank Information for Zero-Shot Detection of Machine-Generated Text

**Jinyan Su[1,2], Terry Yue Zhuo[3], Di Wang[4], Preslav Nakov[1]**

[1]Mohamed bin Zayed University of Artificial Intelligence, [2] Cornell Unversity
[3]Monash University and CSIRO's Data61
[4]King Abdullah University of Science and Technology
`{Jinyan.Su, preslav.nakov}@mbzuai.ac.ae`
`terry.zhuo@monash.edu,di.wang@kaust.edu.sa`

## Abstract

With the rapid progress of Large language models (LLMs) and the huge amount of text they generate, it becomes impractical to manually distinguish whether a text is machine-generated. The growing use of LLMs in social media and education, prompts us to develop methods to detect machine-generated text, preventing malicious use such as plagiarism, misinformation, and propaganda. In this paper, we introduce two novel zero-shot methods for detecting machine-generated text by leveraging the Log-Rank information. One is called DetectLLM-LRR, which is fast and efficient, and the other is called DetectLLM-NPR, which is more accurate, but slower due to the need for perturbations. Our experiments on three datasets and seven language models show that our proposed methods improve over the state of the art by 3.9 and 1.75 AUROC points absolute. Moreover, DetectLLM-NPR needs fewer perturbations than previous work to achieve the same level of performance, which makes it more practical for real-world use. We also investigate the efficiency-performance trade-off based on users' preference for these two measures and provide intuition for using them in practice effectively. We release the data and the code of both methods in `https://github.com/mbzuai-nlp/DetectLLM`.

## 1 Introduction

Large language models (LLMs) have made rapid advancements in recent years, and are now able to generate text with significantly improved diversity, fluency, and quality. Models such as Chat-GPT (OpenAI, 2022), GPT-3 (Brown et al., 2020), LLaMa (Touvron et al., 2023) and BLOOM (Scao et al., 2022) demonstrate exceptional performance in answering questions (Robinson et al., 2022), writing stories (Fan et al., 2018; Yuan et al., 2022) and thus facilitating daily life and improving work efficiency. However, LLMs can also be misused for generating plagiarized text, misinformation, and propaganda, which can lead to negative consequences (Zhuo et al., 2023). For instance, students might use LLMs to write assignments (Rosenblatt, 2023), making fair evaluation difficult for teachers, and in the long run, undermining the integrity of the entire education system. Malicious actors might generate fake news articles to spread misinformation and propaganda or to manipulate public opinion, which is dangerous, especially when it comes to politics (Floridi and Chiriatti, 2020; Stokel-Walker, 2022).

With the proliferation of LLMs and the increasing amount of texts they produce, it is challenging for humans to accurately identify machine-generated texts (Gehrmann et al., 2019). Moreover, it is unrealistic to hire humans to manually identify machine-generated text at scale due to the prohibitively high costs and the efficiency requirements in real-time applications, e.g., in social media. Thus, it is essential to develop tools and strategies to automatically identify machine-generated text and to mitigate the potential negative impact of LLMs.

The problem of distinguishing machine-generated from human-written text is commonly formulated as a binary task (Jawahar et al., 2020). Most previous work has focused on the black-box scenario, where the detector has access to the output of the LLMs only and cannot make use of its internals. Such methods lack flexibility since they need to be retrained from scratch to be able to recognize the output of a new LLM (Mitchell et al., 2023). Given the speed at which new LLMs are developed, black-box methods are becoming more and more expensive and impractical. In cases when the access to the LLM is via an API only, one possibility is for the LLM owner to record all content it has generated, or to watermark all texts it has generated (Kirchenbauer et al., 2023; Zhao et al., 2023). However, such solutions are not feasible for third parties.

We therefore consider a white-box setting, where the detector has full access to the LLMs. We focus on zero-shot methods, where we use the LLM without additional training. Generally speaking, zero-shot methods use the source LLM to extract statistics such as the average per-token log probability or the average rank of each token in the ranked list of possible choices to make a prediction by comparing it to a threshold (Solaiman et al., 2019; Ippolito et al., 2019; Gehrmann et al., 2019; Mitchell et al., 2023). Based on whether the queried statistics are only about the target texts, we can roughly categorize them as perturbation-free and perturbation-based. Perturbation-free methods only query LLMs about the statistics on the target text $x$, while perturbation-based methods such as Mitchell et al. (2023) queries also the statistics of additional perturbed texts, which achieves better performance but is 50-100 times more costly than perturbation-free methods. Thus, there exists a trade-off between performance and efficiency among zero-shot methods.

To mitigate the gap of these two categories and design zero-shot methods with better performance-efficiency balance, we should either improve the accuracy of perturbation-free methods or reduce their cost. Thus, we propose two novel zero-shot methods, one perturbation-free, but more accurate than previous methods, and one perturbation-based method, but with better efficiency.

In the perturbation-free method we proposed, we apply Log-**L**ikelihood Log-**R**ank **r**atio (LRR), which enhances Log-Likelihood information with Log-Rank information as the discerning feature that achieves better performance than existing perturbation-free methods and can even surpass perturbation-based methods on some LLMs. For the perturbation-based method, we use **N**ormalized **p**erturbed log **r**ank (NPR), which is based on the intuition that machine-generated texts are more sensitive to minor rewrites (or small perturbations). Compared to existing perturbation-based methods such as DetectGPT (Mitchell et al., 2023), this approach takes less time and computational resources and is more efficient. We denote these two methods DetectLLM-LRR and DetectLLM-NPR respectively. Our contributions are as follows:

- We propose two novel zero-shot approaches based on Log-Rank statistics, which improve over the state of the art. On average, these methods improved upon the previous best

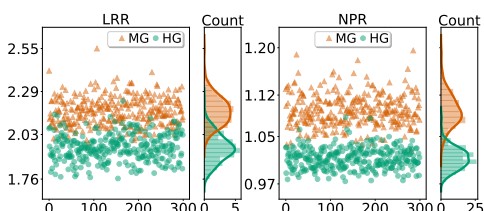

Figure 1: Distribution of LRR and NPR visualized on 300 human-written texts (HG) from the WritingPrompts dataset (Fan et al., 2018) as well as 300 texts generated with GPT-2-xl (MG) by prompting it with the first 30 tokens from human-written texts.

zero-shot methods by 3.9 and 1.75 AUROC points absolute.
- We investigate the efficacy of existing zero-shot methods and explore their limits as the size of the LLMs increases from 1.5 to 20 billion.
- We conduct comprehensive experiments to better understand the efficiency-performance trade-offs in zero-shot methods, thereby providing interesting insights on how to choose among different categories of zero-shot methods based on users' preference for performance or efficiency.

## 2 Related Work

The detection of machine-generated text is commonly formulated as a classification task (Jawahar et al., 2020; Fagni et al., 2021; Bakhtin et al., 2019; Sadasivan et al., 2023; Wang et al., 2023). One way of solving it is to use supervised learning, where a classification model is trained on a dataset containing both machine-generated and human-written texts. For example, GPT-2 Detector (Solaiman et al., 2019) fine-tunes RoBERTa (Liu et al., 2019) on the output of GPT-2, while the ChatGPT Detector (Guo et al., 2023) fine-tunes RoBERTa on the HC3 (Guo et al., 2023) dataset. However, models trained explicitly to detect machine-generated texts may overfit their training distribution of the domains (Bakhtin et al., 2019; Uchendu et al., 2020).

Another stream of work attempts to distinguish machine-generated from human-written texts based on statistical irregularities in the entropy (Lavergne et al., 2008), perplexity (Beresneva, 2016) or in the $n$-gram frequencies (Badaskar et al., 2008). Gehrmann et al. (2019) introduced hand-crafted statistical features to assist humans in detecting-machine generated texts. Moreover, (Solaiman et al., 2019) noted the efficacy of simple zero-shot

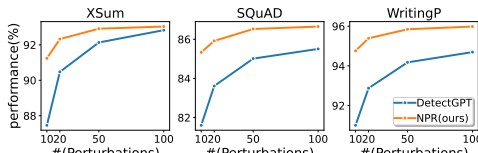

Figure 2: Comparison of DetectGPT to NPR averaged across six models (in terms of AUROC). (The full results are given in Figure 6 in the Appendix).

methods for detecting machine-generated text by evaluating the per-token log probability of texts and using thresholding. Mitchell et al. (2023) observed that machine-generated texts tend to lie in the local curvature of the log probability and proposed DetectGPT, whose prominent performance can only be guaranteed by the large size of the perturbation function and by a large number of perturbations, and thus costs more computational resources.

Other work explored watermarking, which imprints specific patterns of the LLM-output text that can be detected by an algorithm while being imperceptible to humans. Grinbaum and Adomaitis (2022) and Abdelnabi and Fritz (2021) watermarked machine-generated text using syntax tree manipulation, while Kirchenbauer et al. (2023) required access to the LLM's logits at each time step.

# 3 Improved Zero-Shot Approaches by Leveraging Log-Rank Information

In this section, we introduce the Log-**L**ikelihood Log-**R**ank **R**atio (LRR) and the **N**ormalized **P**erturbed log-**R**ank (NPR). LRR combines Log-Rank and Log-Likelihood as they provide complementary information about the text. NPR uses the idea that the Log-Rank of machine-generated texts should be more sensitive to smaller perturbations.

## 3.1 Log-Likelihood Log-Rank Ratio (LRR)

We define the Log-Likelihood Log-Rank Ratio as

$$\text{LRR} = \left| \frac{\frac{1}{t} \sum_{i=1}^{t} \log p_\theta(x_i|x_{<i})}{\frac{1}{t} \sum_{i=1}^{t} \log r_\theta(x_i|x_{<i})} \right|$$
$$= -\frac{\sum_{i=1}^{t} \log p_\theta(x_i|x_{<i})}{\sum_{i=1}^{t} \log r_\theta(x_i|x_{<i})},$$

where $r_\theta(x_i|x_{<i}) \geq 1$ is the rank of token $x_i$ conditioned on the previous tokens.

The Log-Likelihood in the numerator represents the absolute confidence for the correct token, while the Log-Rank in the denominator accounts for

the relative confidence, which reveals complementary information about the texts. As illustrated in Figure 1, LRR is generally larger for machine-generated text, which can be used for distinguishing machine-generated from human-written text. One plausible reason might be that for machine-generated text, the Log-Rank is more discernible than the Log-Likelihood, so LRR illustrates this pattern for machine-generated text. In Sections 4 and 6, we experimentally show that LRR is a better discriminator than either the Log-Likelihood or the Log-Rank. We call the zero-shot method using LRR as a detection feature as DetectLLM-LRR, and use the abbreviation LRR in the rest of the paper.

## 3.2 Normalized Log-Rank Perturbation (NPR)

We define the normalized perturbed Log-Rank as

$$\text{NPR} = \frac{\frac{1}{n} \sum_{p=1}^{n} \log r_\theta(\tilde{x}_p)}{\log r_\theta(x)},$$

where small perturbations are applied on the target text $x$ to produce the perturbed text $\tilde{x}_p$. Here, a perturbation means minor rewrites, such as replacing some of the words. We call the zero-shot method using NPR as a detection feature DetectLLM-NPR, and use the abbreviation NPR in the rest of the paper.

The motivation for NPR is that machine-generated and human-written texts are both negatively affected by small perturbations, i.e., the Log-Rank score will increase after perturbations, but the machine-generated text is more susceptible to perturbations and thus increase more on Log-Rank score after perturbation, which suggests higher NPR score for machine-generated texts. As shown in Figure 1, NPR can be a discernible signal for distinguishing machine-generated from human-written text. DetectGPT (Mitchell et al., 2023) uses a similar idea, but experimentally, we find NPR to be more efficient and to perform better. Details and comparisons are given in Section 4.

# 4 Experimental Setup

In this section, we conduct comprehensive experiments to evaluate the performance of LRR and NPR in comparison to several methods previously proposed in the literature. We experiment with LLM sizes varying from 1.5B to 20B parameters, probing the boundary of zero-shot methods when

LLMs continue to grow in size. We further study the impact of the perturbation function, the number of perturbations (especially for NPR and Detect-GPT), the decoding strategy, and the temperature.

**Data**  Following (Mitchell et al., 2023), we use three datasets: XSum (Narayan et al., 2018), SQuAD (Rajpurkar et al., 2016), WritingPrompts (Fan et al., 2018), containing news articles, Wikipedia paragraphs and prompted stories, respectively, as human-written texts and we produce machine-generated texts using LLMs. These datasets are chosen to represent the areas where LLMs could have a negative impact. For each experiment, we evaluate 300 machine-generated and human-written texts pairs by prompting the LLMs with the first 30 tokens of the human-written text. We release the code for this.

**Evaluation Measure**  Following previous work (Mitchell et al., 2023; He et al., 2023; Krishna et al., 2023), we use the area under the receiver operating characteristic curve (AUROC), which is the probability that a classifier correctly ranks the machine-generated example higher than human-written example. Since for zero-shot methods, detection rates are heavily dependent on the threshold when using discriminative statistics, AUROC is commonly used to measure zero-shot detector performance, which considers the range of all possible thresholds (Krishna et al., 2023).

### 4.1  Methods

**Zero-Shot Methods**  We compare the following:
- $\log p(x)$**:** the idea is that a passage with a high average log probability is more likely to have been generated by the target LLM;
- **Rank:** the idea is that a passage with a higher average rank is more likely to have been generated by the target LLM;
- **Log-Rank:** passage with a higher average observed Log-Rank is more likely to have been generated by the target LLM;
- **Entropy:** machine-generated text has higher entropy;
- **DetectGPT:** machine-generated text has more negative log probability curvature.

More detail and exact definitions of these methods can be found in Appendix A.

These zero-shot baselines, along with our newly proposed LRR and NPR, can be categorized as
- **Perturbation-free**: $\log p(x)$, Rank, Log-Rank, Entropy, LRR. They only query the

LLM for statistics about the target text $x$.
- **Perturbation Based**: DetectGPT and NPR. These methods query the LLM not only for the target text $x$, but also for perturbed versions thereof $\tilde{x}_1, \cdots, \tilde{x}_p$.

As perturbation-based methods perform better (but are also more time-consuming), for fair comparison, we compare them within their own group.

**Supervised Methods**  We also experiment with two supervised detectors: RoBERTa-base and RoBERTa-large. As these are not central to our narrative, we put the results in Appendix B.

**Experimental Details**  For the perturbation-based methods (DetectGPT and NPR), we use T5-3B for perturbation and we perturb the input text 50 times for all the experiments, unless specified otherwise. For all zero-shot methods, we use sampling with a temperature of 1, unless specified otherwise. More detail are given in Appendix A.

## 5  Evaluation Results

### 5.1  Zero-Shot Results

Table 1 shows a comparison of the five baseline zero-shot approaches to our proposed LRR and NPR, grouped as perturbation-based and perturbation-free. We can see that for the perturbation-based methods, NPR consistently outperforms DetectGPT on all datasets and LLMs, except for one case, with an average improvement of 0.90, 2.03, 2.32 AUROC points absolute on XSum, SQuAD, and WritingPrompts, respectively, (using the same perturbation function and the same number of perturbations). For the experiments among perturbation-free methods, on average, our method achieves the best performance and improves by 2.15, 8.27, 1.28 AUROC points absolute over the second-best perturbation-free method (i.e., Log-Rank) on XSum, SQuAD, and WritingPrompts, respectively. Moreover, we find that in some cases, LRR can even perform better than perturbation-based methods, e.g., on SQuAD, LRR outperforms DetectGPT by 4.23 AUROC point absolute and outperforms NPR by 2.20 AUROC points.

### 5.2  Comparing DetectGPT to NPR

Equipped with large perturbation functions and an adequate amount of perturbations, perturbation-based methods generally outperform perturbation-free ones, e.g., using T5-3b as the perturbation function and perturb 50 times as in Table 1. However, in

| Dataset | Perturbation | Method | GPT-2-xl | Neo-2.7 | OPT-2.7 | GPT-j | OPT-13 | Llama-13 | NeoX | Avg. |
|---|---|---|---|---|---|---|---|---|---|---|
| XSum | w/o | log $p$ | 89.16 | 87.69 | 86.98 | 83.10 | 83.90 | 56.89 | 78.16 | 80.84 |
| | | Rank | 79.79 | 77.87 | 76.07 | 76.28 | 74.10 | 48.81 | 72.44 | 72.19 |
| | | Log-Rank | 91.75 | 90.79 | **89.18** | 86.42 | **85.88** | 61.33 | 81.44 | 83.83 |
| | | Entropy | 56.78 | 55.14 | 50.34 | 55.51 | 50.98 | 69.43 | 60.84 | 57.00 |
| | | LRR (ours) | **93.47** | **92.24** | 88.70 | **88.68** | 83.79 | **71.07** | **83.89** | **85.98** |
| | w/ | DetectGPT | 98.80 | 99.11 | 96.02 | **95.88** | 92.65 | 73.55 | 93.58 | 92.80 |
| | | NPR (ours) | **99.40** | **99.46** | **97.09** | 95.76 | **94.63** | **75.51** | **94.08** | **93.70** |
| SQuAD | w/o | log $p$ | 90.72 | 84.18 | 87.84 | 78.20 | 80.65 | 42.91 | 68.78 | 76.18 |
| | | Rank | 83.46 | 79.77 | 81.85 | 79.46 | 77.47 | 54.44 | 73.10 | 75.65 |
| | | Log-Rank | 94.33 | 89.52 | 91.76 | 83.37 | 85.05 | 48.28 | 73.88 | 80.88 |
| | | Entropy | 57.97 | 58.48 | 53.29 | 58.26 | 57.14 | **69.71** | 59.97 | 59.26 |
| | | LRR (ours) | **97.42** | **95.74** | **95.89** | **91.59** | **91.36** | 68.78 | **83.31** | **89.15** |
| | w/ | DetectGPT | 98.52 | 95.86 | 96.91 | 88.66 | 90.60 | 47.03 | 76.84 | 84.92 |
| | | NPR (ours) | **99.40** | **97.56** | **98.39** | **91.88** | **93.04** | **48.67** | **79.73** | **86.95** |
| WritingP | w/o | log $p$ | 96.71 | 95.63 | 95.05 | 94.43 | 92.53 | 83.54 | 93.27 | 93.02 |
| | | Rank | 87.62 | 82.79 | 83.89 | 83.21 | 83.52 | 77.64 | 81.64 | 82.90 |
| | | Log-Rank | 98.02 | 97.15 | 96.32 | 96.06 | 94.34 | 88.11 | 95.14 | 95.02 |
| | | Entropy | 36.45 | 34.07 | 39.75 | 36.93 | 42.49 | 47.64 | 37.89 | 39.32 |
| | | LRR (ours) | **98.34** | **98.02** | 96.45 | **96.97** | **95.09** | **92.66** | **96.56** | **96.30** |
| | w/ | DetectGPT | 99.30 | 98.71 | 98.33 | 95.52 | 96.46 | 83.01 | 92.94 | 94.90 |
| | | NPR (ours) | **99.78** | **99.59** | **98.87** | **98.07** | **98.14** | **89.39** | **96.72** | **97.22** |

Table 1: **Zero-shot experiments.** Comparison of the proposed LRR and NPR to other zero-shot methods in terms of AUROC. For fair comparison, we show in bold the best results, both with and without perturbations.

| Decoding | Dataset | w/o Perturbation | | | | | w/ Perturbation | |
|---|---|---|---|---|---|---|---|---|
| | | log $p$ | Rank | Log-Rank | Entropy | LRR (ours) | DetectGPT | NPR (ours) |
| top-$k$ | XSum | 81.64 | 70.68 | 85.19 | 55.47 | **89.25** | 91.34 | **92.93** |
| | SQuAD | 76.31 | 74.31 | 81.28 | 57.96 | **90.61** | 82.42 | **84.99** |
| | WritingP | 93.80 | 82.15 | 95.72 | 37.26 | **97.10** | 93.89 | **96.33** |
| top-$p$ | XSum | 86.94 | 70.86 | **88.65** | 53.89 | 88.29 | 92.74 | **93.42** |
| | SQuAD | 82.07 | 75.03 | 85.49 | 55.86 | **91.09** | 83.98 | **86.19** |
| | WritingP | 96.51 | 82.48 | **97.44** | 33.92 | 97.25 | 94.20 | **96.55** |

Table 2: **Decoding strategy analysis.** Shown are the AUROC scores for methods with top-$k$ ($k = 40$) and top-$p$ ($p = 0.96$) sampling averaged across four LLMs: Neo-2.7, OPT-2.7, GPT-j, Llama-13.

practice, due to time and resource constraints, not all users can afford these models and large amounts of perturbations. Thus, it is important to investigate how NPR and DetectGPT behave with smaller perturbation function size and fewer perturbations.

**Different Number of Perturbations.** Figure 2 shows the averaged performance of DetectGPT and NPR with varying number of perturbations. We can see that NPR consistently performs better than DetectGPT when using the same number of perturbations. In other words, NPR can achieve a comparable or better performance but with significantly fewer perturbations. For example, in SQuAD and WritingPrompts dataset, NPR achieves 85 points and 95 points using approximately 10 perturbations while DetectGPT requires around 100 perturbations, which highlights the effectiveness and efficiency of NPR. More complete results for each dataset and model can be found in Figure 6 and Figure 7 of Appendix C.

**Different Perturbation Functions.** In Table 3, we compare NPR to DetectGPT using a smaller perturbation model T5-large, and the result is averaged over 6 LLMs and 3 datasets. We found

that replacing T5-3b with smaller models harms the performance of both NPR and DetectGPT, and the performance degradation can't be mitigated by increasing the number of perturbations. For both NPR and DetectGPT, the average performance of 100 perturbations with T5-large is still worse than 10 perturbations with T5-3b (emphasized with the grey box in Table 3). Moreover, one can observe that, NPR is less affected by the reduced perturbation function size: when replacing T5-3b to T5-large, the performance degradation averaged over 10, 20, 50, 100 perturbations for NPR is 4.40 points, much smaller compared to that of 8.06 points for DetectGPT. The complete results on 6 LLMs and 3 datasets can be found in Figure 8 of Appendix C.

### 5.3 Different Decoding Strategy and Temperature

**Alternative Decoding Strategies.** In line with prior work (Pagnoni et al., 2022), we experimented with top-$k$ sampling (Fan et al., 2018) and top-$p$ sampling (Holtzman et al., 2019). Top-$k$ sampling generates from top-$k$ most likely words according to the LLM. Top-$p$ sampling (nucleus sampling) samples from the set of words that collectively ac-

| Perturbation Function | Dataset | # (Perturbations) | | | |
|---|---|---|---|---|---|
| | | 10 | 20 | 50 | 100 |
| NPR (ours) | T5-large | 86.69 | 88.00 | 88.74 | 88.94 |
| | T5-3b | 91.39 | 92.35 | 93.04 | 93.20 |
| | Diff | **4.70** | **4.35** | **4.30** | **4.26** |
| DetectGPT | T5-large | 77.94 | 81.12 | 83.90 | 84.54 |
| | T5-3b | 86.70 | 89.57 | 91.38 | 92.10 |
| | Diff | 8.76 | 8.45 | 7.48 | 7.56 |

Table 3: **Perturbation analysis.** Comparing DetectGPT to NPR using different perturbations (AUROC scores).

count for a total mass probability $p$. The results (averaged across 4 LLMs) are shown in Table 2, and complete results can be found in Table 9 of Appendix D. We find that, although almost all the zero-shot methods perform better when using top-$k$ and top-$p$ sampling than temperature sampling, Log-Rank and Log-Likelihood methods are more in favour of top-$p$ sampling, while LRR is stable in both top-$p$ and top-$k$ sampling. For top-$k$ decoding, LRR improves 4.06, 9.33, and 1.38 points over the second-best zero-shot method baseline on three datasets, respectively. LRR performance also improves when using top-$p$ decoding strategy, but due to the unstable performance surge of the Log-Rank method, LRR become slightly behind the Log-Rank method, with a minor difference of 0.36 and 0.19 points on the XSum and WritingPrompts datasets, respectively. For perturbation-based methods, their behaviour is consistent with previous results, where NPR outperforms DetectGPT for both top-$p$ and top-$k$ sampling strategies.

**Different Temperature.** Temperature controls the degree of randomness of the generation process. Increasing the temperature leads to more randomness and creativity while reducing it leads to more conservation and less novelty. In practice, people adjust the temperature for their specific purposes. For example, students might set a high temperature to encourage more original and diverse output when writing a creative essay, whereas fake news producers might set lower temperatures to generate seemingly convincing news articles for their deceptive purposes. Based on our experiments in Table 4, we found that Log-Likelihood ($\log p$), Log-Rank and LRR is highly sensitive to the temperature and can get even better results than perturbation-based methods when the temperature is relatively low. In addition, the performance improvement of the Rank method with the increased temperature is negligible compared to Log-Likelihood, Log-Rank and LRR, while the performance of the entropy method seems to be positively correlated to the temperature. We conjure that the abnormal behaviour of

the Entropy method might be because of the assumption that "machine-generated text has higher entropy" (Mitchell et al., 2023), which, from our experiments, doesn't stand for high temperature. As for the perturbation-based method, the impact of temperature is not as clear as a perturbation-free method. But in general, the results suggest the temperature has only minor effects on DetectGPT while it improves the performance of NPR. Another observation is that the perturbation-free method performs better than the perturbation-based method in low temperatures, for example, if the temperature is smaller than 0.95, perturbation-based methods get better detection accuracy while being efficient.

# 6 Analysis of the Efficiency

Though in Table 1, perturbation-based methods appear to be significantly better than perturbation-free methods, it is important to note that their superior performances can only be achieved with large perturbation functions and multiple number of perturbations, which leads to intensive demand for computational resources and longer computational time. Thus, while performance is an important factor, it is crucial to consider the efficiency of these zero-shot methods as well.

## 6.1 Computational Cost Analysis

To get an idea of how costly different zero-shot methods are to achieve their performance in Table 1, we estimated the computational time (per sample) for each zero-shot method in Table 5. The time is estimated over the average of 10 samples. For perturbation-based methods, since the time depends on the perturbation function and the number of perturbations, we used T5-3b as the perturbation function and use 50 perturbations since this is the setting used for the main results in Table 1, we want to provide an idea of how much more it costs for perturbation based method to achieve exceptional performance in Table 1. We observed that the computational time of Log-Likelihood, Rank, Log-Rank and Entropy are almost the same, while LRR runs approximately 2 times longer than these methods since it requests both the Log-Rank and Log-Likelihood statistics. For perturbation-based methods, the running time is at least 50 times longer compared to Log-Likelihood, Rank, Log-Rank, and Entropy method, since they calculate the Log-Likelihood or Log-Rank for not only the target text but also perturbed samples.

| | | w/o Perturbation | | | | w/ Perturbation | |
|---|---|---|---|---|---|---|---|
| Temperature | $\log p$ | Rank | Log-Rank | Entropy | LRR (ours) | DetectGPT | NPR (ours) |
| 0.5 | 98.72 | 77.87 | **99.29** | 25.90 | 99.23 | 86.14 | **95.76** |
| 0.7 | 97.01 | 76.98 | 98.05 | 38.28 | **98.84** | 90.28 | **95.61** |
| 0.9 | 90.04 | 75.82 | 92.28 | 47.14 | **94.50** | 90.33 | **92.89** |
| 0.95 | 86.15 | 75.43 | 88.88 | 50.42 | **92.04** | 89.97 | **91.98** |
| 1 | 81.48 | 74.85 | 84.81 | 52.37 | **89.15** | 89.02 | **90.86** |

Table 4: **Temperature experiments.** Results of using different temperatures (AUROC scores).

| Perturbation | Method | GPT-2-xl | Neo-2.7 | OPT-2.7 | GPT-j | OPT-13 | Llama-13 | NeoX |
|---|---|---|---|---|---|---|---|---|
| w/o | $\log p$ | 0.06 | 0.09 | 0.10 | 0.04 | 0.07 | 0.07 | 0.60 |
| | Rank | 0.07 | 0.10 | 0.09 | 0.04 | 0.05 | 0.07 | 0.60 |
| | Log-Rank | 0.06 | 0.09 | 0.10 | 0.04 | 0.05 | 0.06 | 0.60 |
| | Entropy | 0.06 | 0.09 | 0.09 | 0.04 | 0.05 | 0.06 | 0.60 |
| | LRR (ours) | 0.12 | 0.19 | 0.18 | 0.08 | 0.10 | 0.14 | 1.20 |
| w/ | DetectGPT | 8.07 | 9.60 | 9.80 | 7.03 | 7.98 | 8.14 | 35.56 |
| | NPR (ours) | 8.15 | 9.69 | 9.90 | 7.12 | 7.83 | 7.98 | 35.67 |

Table 5: Computational time (seconds) for different zero-shot methods on different LLMs (averaged over 10 reruns).

| $t_p(s)$ | | | | $t_m(s)$ | | | | | | |
|---|---|---|---|---|---|---|---|---|---|---|
| T5-3b | T5-large | T5-base | T5-small | GPT-2-xl | Neo-2.7 | OPT-2.7 | GPT-j | OPT-13 | Llama-13 | NeoX |
| 0.10 | 0.08 | 0.04 | 0.03 | 0.06 | 0.09 | 0.10 | 0.04 | 0.07 | 0.07 | 0.60 |

Table 6: **Computation time.** Estimated computation time for one perturbation ($t_p$) and for calculating the target statistics on the text ($t_m$): shown in seconds.

**Composition of the Computational Time.** In general, for perturbation-free zero-shot methods, the computational time only depends on the size of LLM and the complexity of statistics. LRR is twice as complex as simple statistics such as Log-Rank and Log-Likelihood, so it takes approximately twice as long to compute. As for LLM size, intuitively, larger models usually take more time to compute, which can also be observed in Table 5. The additional computational time of perturbation-based methods comes from two folds: (1) The total time for perturbation, which depends on the perturbation function we use and the number of perturbations. (2) The total time for calculating statistics of the perturbed texts, which depends on the number of perturbations, the size of LLM and the complexity of statistics. To reduce the computational time of the perturbation-based method, we could either choose a smaller size of the perturbation function or reduce the number of perturbations.

**Formula for Estimating the Computational Time.** Let $t_p$ be the time of perturbing 1 sample, $t_m$ be the time of calculating simple statistics (such as Log-Likelihood) of one sample for a particular LLM and $n$ be the number of perturbations. The computational time for Log-Likelihood, rank, Log-Rank, and entropy is approximately $t_m$, the estimated time for LRR is $2 \cdot t_m$, while the estimated computational time for the perturbation-based method is $n \cdot t_p + (n+1) \cdot t_m$. The estimated

values of $t_p$ and $t_m$ are illustrated in Table 6, which can help us estimate the total running time (in seconds) of different zero-shot methods.

## 6.2 Balancing Efficiency and Performance

In this subsection, we provide additional experiments on LRR (the best perturbation-free method) and NPR (the best perturbation-based method, more time-consuming than LRR but also rather satisfactory performance) to provide users with some intuition on setting parameters of NPR and choosing among between these two methods according to user's preference of efficiency and performance.

First, we study the perturbation function used for NPR. Different from Section 5.2, where the focus is to illustrate the advanced performance of NPR compared with DetectGPT, here, we mainly focus on the efficiency performance trade-off perspective and provide some intuition on choosing perturbation functions.

**T5-small and T5-base are not good candidates for perturbation functions.** T5-small and T5-base are 2 or 3 times faster than larger models such as T5-large (as shown in Table 6), one might wonder if it is possible to trade the saved time with more perturbations for a better performance? We give a negative answer to this. We observe in Figure 3 that using T5-base and T5-small performs worse than LRR even with 50 to 100 perturbations, which suggests that LRR can be at least 50 to 100 times faster

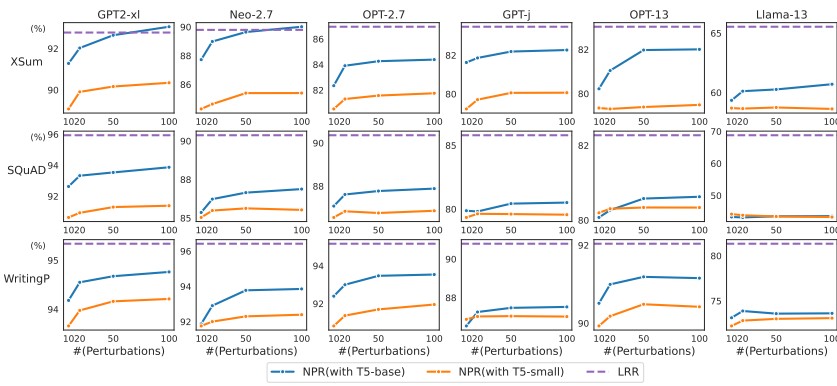

Figure 3: Comparing LRR and NPR when T5-small and T5-base are used for perturbation in NPR (AUROC scores).

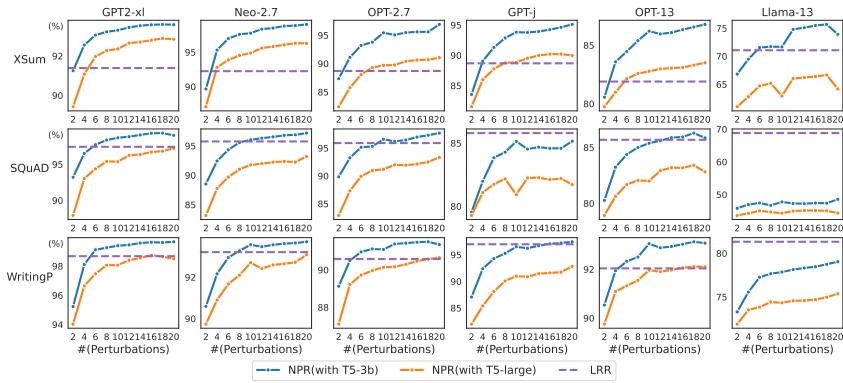

Figure 4: Comparing LRR and NPR using T5-3b and T5-large for perturbation in NPR (AUROC scores).

while outperforming perturbation based methods. So, if the user can only afford T5-small or T5-base as a perturbation function, they should choose LRR with no hesitation since it achieves both better efficiency and better performance.

**Cost-Effectiveness on More Perturbations and Larger Perturbation Function.** In Figure 4, we illustrate the effectiveness of LRR compared to NPR with T5-large and T5-3b as perturbation function respectively, from which, we find that (1) T5-3b has a higher performance upper limits compared with T5-large. So, if resources are allowed (enough memory and adequate perturbation time), t5-3b would be a better choice, especially for users who prioritize performance. (2) To achieve the same performance as LRR, generally, we only need less than 10 perturbations using T5-3b as the perturbation function. This estimate could help us choose whether to use NPR or LRR on the validation set: setting the number of perturbations to be 10, if LRR outperforms NPR, we would suggest using LRR, otherwise, NPR would be a better option. (3) To achieve the same performance, using T5-large takes more than 2 times perturbations than using

T5-3b, while the perturbation time using T5-3b is less than twice the time using T5-large, so using large perturbation functions such as T5-3b is much more efficient than using smaller ones such as T5-large. The only concern is the memory.

In summary, we suggest using the larger perturbation functions if memory permits, which is more cost-effective: and less time-consuming for the same performance and has a high-performance upper limit. Moreover, setting the number of perturbations to 10 would be a good threshold on the validation set to decide whether to use NPR or LRR.

# 7 Conclusion

In this paper, we proposed two simple but effective zero-shot machine-generated text detection methods by leveraging the Log-Rank information. The methods we proposed —LRR and NPR—, achieve state-of-the-art performance within their respective category. In addition, we explored different settings such as decoding strategy and temperatures, as well as different perturbation functions and number of perturbations to better understand the advantages

and the disadvantages of different zero-shot methods. Then, we analyzed the computational costs of these methods, and we provided guidance on balancing efficiency and performance.

## Limitations

One of the limitations of zero-shot methods is the white box assumption that we can have some statistics about the source model. This induces two problems: for closed-source models (such as GPT-3), these statistics might not have been provided. Moreover, in practice, the detector might have to run the model locally to get the statistics for the purpose of detection, which requires that the detector have enough resources to use the LLM for inference. Based on the limitations of zero-shot methods, we consider weakly supervised learning (Ratner et al., 2017) as an important direction for future work. Though many papers in detecting machine-generated text assume knowing the source LLM where the text is generated from, in realistic, the source LLM might be unknown, so it is worth combining weak supervised learning as well as weak supervision sources (other LLMs at hand that might not be the target LLM) to weakly train a classifier. With the flexibility of the weak supervision sources, the limitations of our work could possibly be addressed: (1) Since the weak supervision sources do not have to be from the same target model, there is no need to assume that the target LLM is known. (2) Since the weak supervision sources are classifiers, we could only use statistics that are within reach, or even statistics from other open-source LLMs. (3) The weak supervision sources can be from smaller LLMs, rather than the target LLM, this relaxes the requirement for running an extremely large LLM locally.

In addition, our conclusions hugely rely on our English-centric experiments. It is worth noting that the detection of machine-generation text in other languages is also important, especially for low-source languages. We encourage future studies on the investigation of zero-shot detection in multilingual settings.

## Ethics and Broader Impact

Although our paper focuses on the malicious use of LLMs such as spreading misinformation and propaganda, or dishonesty in the education system, it's necessary to recognize that LLMs also have a wide range of potential benefits, and we would like to point out that most of the people apply LLMs for good conducts such as improving their work efficiency.

Moreover, even though our detectors achieve high AUC scores, it should be recognized that every machine-generated text detector, including ours, has its limitations and can make mistakes, we can't guarantee 100% accuracy for every sample. As such, when deciding whether a text, such as a student's essay, is written by a human or machine, our results are for reference only and should not be used as concrete evidence for punishment. For ethical concerns, our detector should only assist humans to make decisions, rather than directly make decisions for users, thus, we recommend users take these results as one of many pieces in a holistic assessment of texts.

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

# A  Experimental Details and Baselines

**Details on Baselines.**  We mainly compare the proposed methods with zero-shot methods, which utilize the source model itself to extract distinguishable statistic features, including:

- Log-Likelihood ($\log p$) (Solaiman et al., 2019): This approach evaluates the average token-wise log probability of the text and classifies text with higher Log-Likelihood to be machine-generated.

- Rank (Gehrmann et al., 2019): This approach evaluates the average rank of each token of the text and classifies text with a smaller average rank to be machine-generated.

- Log-Rank (Mitchell et al., 2023): Instead of using the Rank score directly, this approach evaluates the average Log-Rank of each token of the text and classifies text with a smaller average Log-Rank to be machine-generated.

- Entropy (Gehrmann et al., 2019): This approach is inspired by the hypothesis that machine-generated texts are more likely to have over-confident (thus low entropy) predictive distributions. In practice, (Mitchell et al., 2023) discovered that entropy is positively correlated with passage fakeness, therefore, following their convention, we use high average entropy as a signal of machine-generated text.

- DetectGPT (Mitchell et al., 2023): DetectGPT is based on the hypothesis that when applying small perturbations to a passage $x$ and produce the perturbed text $\tilde{x}$, the quantity $\log p_\theta(x) - \log p_\theta(\tilde{x})$ is relatively larger for machine-generated samples than human written one. In practice, the performance of this approach depends heavily on the external perturbation function and the number of perturbations.

**Details on LLMs used.**  We used 7 LLMs ranging from 1.5B parameters to 20B parameters in our main experiments.

- GPT-2-xl (Radford et al., 2019) is the 1.5B parameter version of GPT-2 trained on a dataset of 8 million web pages called WebText (Radford et al., 2019), whose objective is to predict the next word given previous words within the text. GPT-2-xl surpasses many other language models trained on specific domains (such as books, news, Wikipedia) without using domain-specific training datasets.

- GPT-Neo-2.7B (Black et al., 2021) was trained as an autoregressive language model on Pile (Gao et al., 2020) dataset with EleutherAI's replication of the GPT-3 architecture.

- OPT-2.7B and OPT-13B are two models among a collection of decoder-only pre-trained transformers introduced in (Zhang et al., 2022), with the performance roughly matching GPT-3 of the same size.

- GPT-j-6B (Wang and Komatsuzaki, 2021), which was also trained on Pile (Gao et al., 2020), exhibits zero-shot performance roughly comparable to GPT-3 of comparable size. In addition, the performance gap from GPT-3 of similar size is closer than the GPT-Neo models.

- Llama-13b is the 13B parameter model from Llama models (Touvron et al., 2023): a collection of models ranging from 7B to 65B parameters trained with publicly available datasets. Llama-13B outperforms GPT-3 (175B) on most benchmarks, and all the models are released to the research community.

- NeoX-20B (Black et al., 2022) is a 20B autoregressive model trained on Pile, whose weights have been released openly to the public.

**Experimental Details.**  For small models such as GPT-2-xl, Neo-2.7, OPT-2.7, GPT-j, we use 1 NVIDIA A100 GPU (with total memory 40G) in our experiments; for larger models such as OPT-13b and Llama-13, we use 3 A100 GPUs (total memory 120 G) while using 4 A100 GPUs (total memory 160 G) for the largest model NeoX-20.

|  |  | GPT-2-xl | Neo-2.7 | OPT-2.7 | GPT-j | OPT-13 | Llama-13 | NeoX | Avg. |
|---|---|---|---|---|---|---|---|---|---|
| XSum | RoBERTa-base | 97.57 | 96.82 | 94.86 | 90.37 | 88.62 | 79.18 | 88.96 | 90.91 |
|  | RoBERTa-large | **99.74** | **99.73** | **98.37** | **97.58** | **93.85** | **85.93** | **95.13** | **95.76** |
| SQuAD | RoBERTa-base | 97.65 | 94.42 | 92.56 | 87.57 | 88.96 | 76.98 | 84.37 | 88.93 |
|  | RoBERTa-large | **99.01** | **98.30** | **96.53** | **93.31** | **91.62** | **82.59** | **88.37** | **92.82** |
| WritingP | RoBERTa-base | 96.88 | 95.23 | 89.57 | 93.26 | 86.18 | 83.49 | 88.92 | 90.50 |
|  | RoBERTa-large | **98.75** | **98.80** | **94.98** | **97.11** | **88.75** | **88.32** | **93.72** | **94.35** |

Table 7: Complete results for the supervised methods (AUROC score).

|  |  | top-$k$ | | | | top-$p$ | | | |
|---|---|---|---|---|---|---|---|---|---|
|  |  | Neo-2.7 | OPT-2.7 | GPT-j | Llama-13 | Neo-2.7 | OPT-2.7 | GPT-j | Llama-13 |
| XSum | RoBERTa-base | 96.48 | 94.15 | 92.72 | 82.47 | 98.30 | 97.30 | 96.71 | 85.84 |
|  | RoBERTa-large | **99.74** | **98.06** | **98.29** | **87.33** | **99.84** | **98.97** | **98.82** | **89.35** |
| SQuAD | RoBERTa-base | 93.55 | 93.27 | 87.60 | 76.79 | 96.34 | 97.65 | 92.26 | 84.05 |
|  | RoBERTa-large | **98.35** | **96.88** | **93.97** | **82.42** | **98.21** | **98.39** | **95.09** | **86.46** |
| WritingP | RoBERTa-base | 97.27 | 90.14 | 93.86 | 83.24 | 98.33 | 94.09 | 96.55 | 88.78 |
|  | RoBERTa-large | **99.34** | **96.34** | **97.12** | **87.05** | **99.68** | **96.06** | **97.94** | **89.59** |

Table 8: Complete results for the supervised methods using top-$k$ ($k = 40$) and top-$p$ ($p = 0.96$) sampling across four models (AUROC scores).

## B    Supervised Methods

**Main results for supervised methods.**    Comparing Table 1 with Table 7, we found that, on average, our best zero shot method (either LRR on SQuAD dataset or NPR on XSum and WritingPrompts dataset) can exceed supervised model fine-tuned on RoBERTa-base. For the larger model RoBERTa-large, only on writing dataset, perturbation-based method DetectGPT and NPR outperform RoBERTa-large model, by a margin of 0.55% and 2.87% respectively.

**Supervised Method with Different Decoding Strategy.**    We experimented the 4 models used in zero-shot methods with top-$p$ and top-$k$ decoding strategy for the supervised method and found that using top-$p$ decoding strategy performs better than using top-$k$. (See Table 8). Compared to zero-shot methods, the best zero-shot method NPR can outperform the RoBERTa-base model while being comparable to the RoBERTa-large model.

**Supervised Method with Different Temperature.**    Supervised methods also perform better with lower temperature, but zero-shot methods such as Log-Rank and Log-Likelihood methods might exceed supervised methods in low temperature. Moreover, we found that the performance gap of RoBERTa-base and RoBERTa-large would be narrowed with lower temperature. The results are illustrated in Figure 5.

## C    Comparing NPR and DetectGPT

**Different Number of Perturbations.**    The results for models smaller than or equal to 13B parameters are shown in Figure 6. For the NeoX-20b model, we don't have enough computation resources to perform 100 perturbations, so we show it separately in Figure 7 with 1, 10, 20, and 50 perturbations. For XSum dataset, NPR and DetectGPT almost coverages with 100 perturbations, but for the SQuAD and WritingPrompts dataset, NPR still outperforms DetectGPT even with 100 perturbations. For the SQuAD dataset with the Llama-13b model, DetectGPT exhibits abnormality while NPR maintains stably improved performance as the number of perturbations increases. In addition, in nearly all the datasets and models, NPR outperforms DetectGPT except GPT-j on the XSum dataset, demonstrating the effectiveness of NPR compared to DetectGPT.

**Using T5-large as Perturbation Function.**    We illustrate the performance of NPR and DetectGPT in Figure 8 with different combinations of dataset and LLMs using T5-large as a perturbation function. Compared to T5-3b illustrated in Figure 6, the superiority of NPR over DetectGPT becomes more distinct with T5-large being the perturbation function, where in almost all the LLMs, datasets and different numbers of perturbations (except with Llama-13b on SQuAD), NPR outperforms DetectGPT by a large margin. In addition, we could also observe that NPR achieves comparable or even better results with only

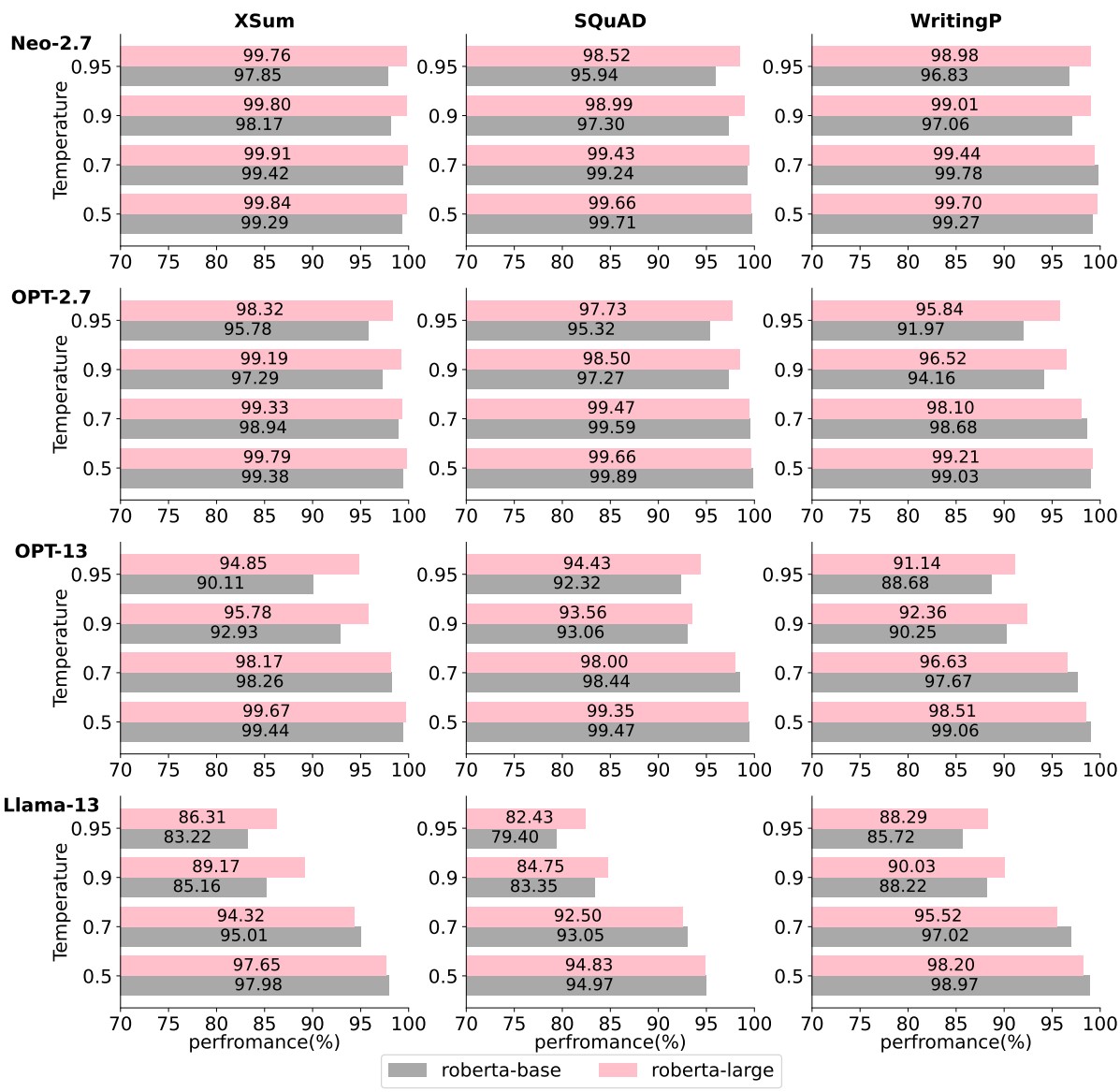

Figure 5: Comparing supervised methods with different temperature (AUROC score).

10 perturbations to that of DetectGPT with 100 perturbations, which indicates that NPR is more efficient and can achieve a similar level of performance with significantly fewer perturbations.

## D  Alternative Sampling Strategies and Temperature

**Different Sampling Strategy.**  In Table 9, we illustrate the complete results with different zero-shot methods with four LLMs using top-$p$ and top-$k$ sampling. For perturbation-based methods, even with different sampling strategies, NPR provides a clearer signal for machine-generated text detection than DetectGPT. Moreover, we find that although LRR is more stable than Log-Rank and Log-Likelihood methods: when replacing temperature sampling to top-$p$ and top-$k$ sampling, all the above-mentioned three zero-shot methods' performance improves, however, LRR improves approximately the same for both top-$k$ and top-$p$ sampling while the other two is more in favour of top-$p$ sampling.

**Different Temperature.**  Here, we investigate how the temperature used for machine-generated texts affects the detection accuracy of different zero-shot methods. From Figure 9, we find that all the perturbation-free zero-shot methods improved their performance with the decreasing temperature. In particular, for the Log-Rank and Log-Likelihood method, the performance can become extremely high

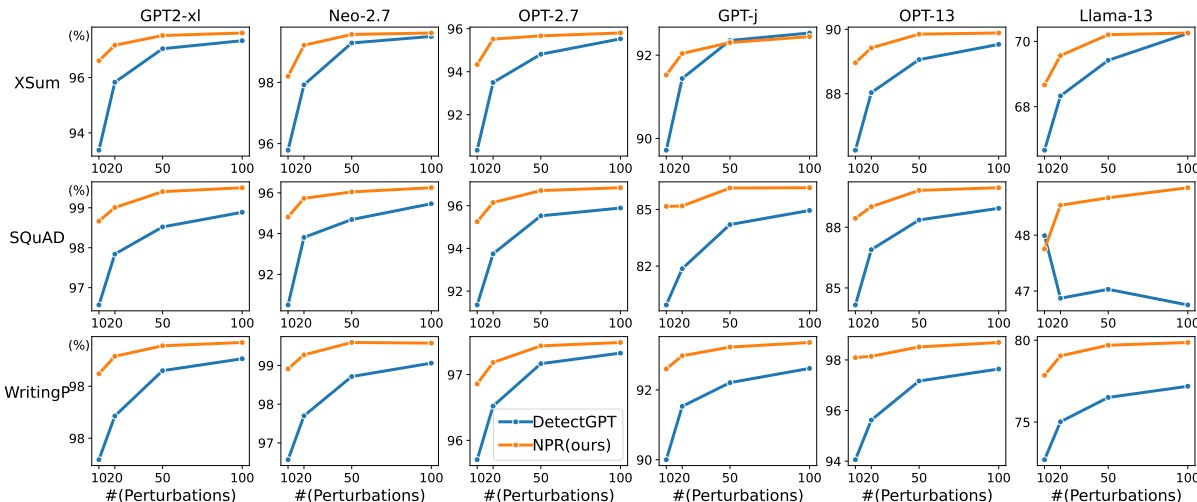

Figure 6: Comparing DetectGPT and NPR (AUROC score).

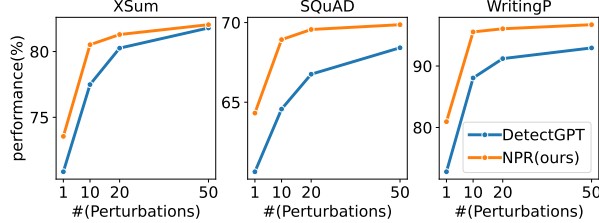

Figure 7: Comparing DetectGPT and NPR on NeoX-20b (AUROC score).

when the temperature drops, even exceeding NPR and achieving approximately 100 points detection accuracy. For example, in Neo-2.7 and OPT-13 with temperature 0.5, $\log p$ method and Log-Rank method achieve an accuracy of 100 points on WritingPrompts dataset, this prevalent performance can be observed notably in smaller models with relatively high temperature (such as GPT-2-xl and Neo-2.7 with high temperature such as 0.7) or in large models with relatively lower temperature such as OPT-13 with temperature 0.5 as we demonstrated in Figure 9. Though we omit the entropy method because it gets an accuracy worse than random guessing, one of the observations from our experiments is that using the assumption "machine-generated text has higher entropy" suggested in (Mitchell et al., 2023), the performance of the entropy method improves with the increasing temperature with absolute accuracy smaller than 50 points, which suggests that for low temperature, we should use the assumption "machine-generated text has lower entropy" for detection machine-generated text. In general, the Entropy method performs worse than random and is not an implementable detection method.

For perturbation-based methods (Figure 10), while DetectGPT does not exhibit a clear trend with respect to temperature, the performance of NPR improves with the decreasing temperature most of the time. However, this trend is not as clear as the Log-Rank and Log-Likelihood methods, especially when the temperature becomes too low. This behaviour suggests that the perturbation-based method is more suitable for high temperatures, while the perturbation-free method is more suitable for low temperature.

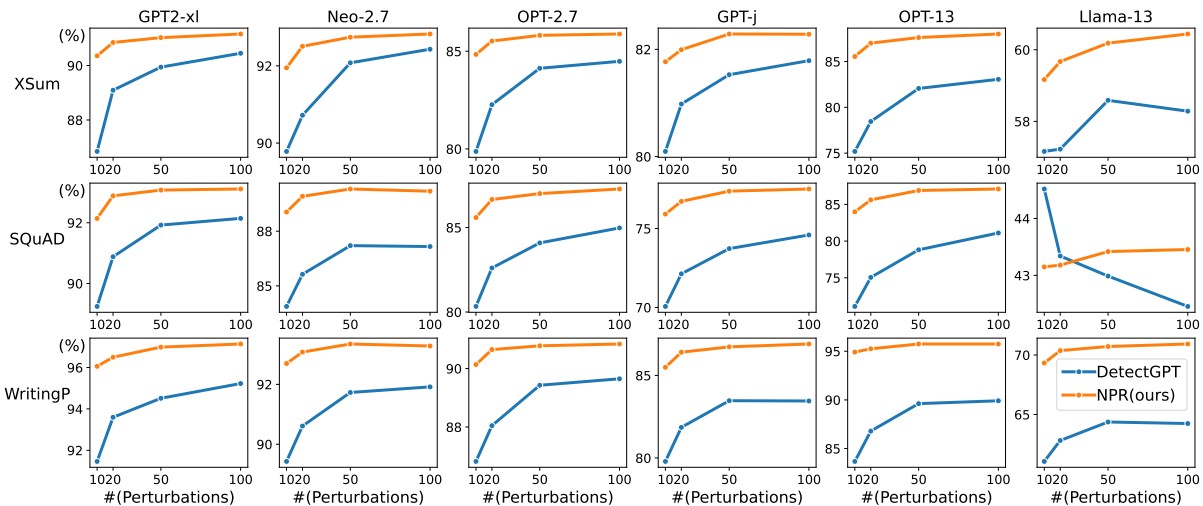

Figure 8: Comparing DetectGPT and NPR using t5-large (AUROC score).

| Dataset | Perturbation | Method | top-$k$ | | | | top-$p$ | | | |
|---|---|---|---|---|---|---|---|---|---|---|
| | | | Neo-2.7 | OPT-2.7 | GPT-j | Llama-13 | Neo-2.7 | OPT-2.7 | GPT-j | Llama-13 |
| XSum | w/o | log $p$ | 91.27 | 90.19 | 85.95 | 59.14 | 95.52 | 93.27 | 91.13 | 67.86 |
| | | Rank | 78.79 | 76.75 | 77.25 | 49.94 | 78.58 | 76.89 | 77.18 | 50.77 |
| | | Log-Rank | 94.20 | 92.30 | 89.18 | 65.09 | **96.71** | **93.93** | **92.53** | 71.44 |
| | | Entropy | 53.07 | 47.80 | 53.23 | 67.76 | 49.05 | 46.41 | 52.16 | 67.94 |
| | | LRR (ours) | **95.50** | **92.35** | **91.14** | **77.99** | 95.64 | 90.68 | 91.14 | **75.72** |
| | w/ | DetectGPT | 98.94 | 96.63 | **96.56** | 73.22 | 98.82 | 97.72 | 96.58 | 77.82 |
| | | NPR (ours) | **99.61** | **98.23** | 96.41 | **77.48** | **99.27** | **98.40** | **97.35** | **78.67** |
| SQuAD | w/o | log $p$ | 87.85 | 91.00 | 81.32 | 45.06 | 91.20 | 94.24 | 86.69 | 56.16 |
| | | Rank | 80.10 | 82.14 | 79.81 | 55.21 | 80.56 | 82.40 | 80.28 | 56.89 |
| | | Log-Rank | 92.58 | 94.40 | 86.94 | 51.21 | 94.48 | 96.37 | 90.44 | 60.66 |
| | | Entropy | 54.62 | 50.83 | 56.89 | 69.52 | 54.51 | 50.01 | 55.67 | 63.26 |
| | | LRR (ours) | **97.79** | **97.58** | **94.55** | **72.52** | **97.48** | **98.11** | **94.38** | **74.38** |
| | w/ | DetectGPT | 97.04 | 97.53 | 87.59 | 47.52 | 97.50 | 97.48 | 88.90 | 52.06 |
| | | NPR (ours) | **98.56** | **99.35** | **91.21** | **50.83** | **98.32** | **99.18** | **92.99** | **54.28** |
| WritingP | w/o | log $p$ | 96.62 | 95.99 | 95.67 | 86.93 | 98.16 | 98.10 | 97.11 | 92.68 |
| | | Rank | 82.67 | 83.96 | 83.49 | 78.49 | 82.89 | 84.45 | 83.55 | 79.01 |
| | | Log-Rank | 97.90 | 97.23 | 97.20 | 90.57 | **98.73** | **98.60** | **97.89** | 94.56 |
| | | Entropy | 32.37 | 38.22 | 34.37 | 44.09 | 27.08 | 36.77 | 32.82 | 39.03 |
| | | LRR (ours) | **98.58** | **97.97** | **98.06** | **93.80** | 98.46 | 97.97 | 97.76 | **94.79** |
| | w/ | DetectGPT | 99.05 | 98.65 | 96.05 | 81.83 | 98.80 | 98.62 | 96.67 | 82.70 |
| | | NPR (ours) | **99.58** | **99.46** | **98.27** | **87.99** | **99.36** | **99.04** | **97.85** | **89.96** |

Table 9: Complete result for the zero-shot methods using top-$k$ and top-$p$ sampling across four models (AUROC score).

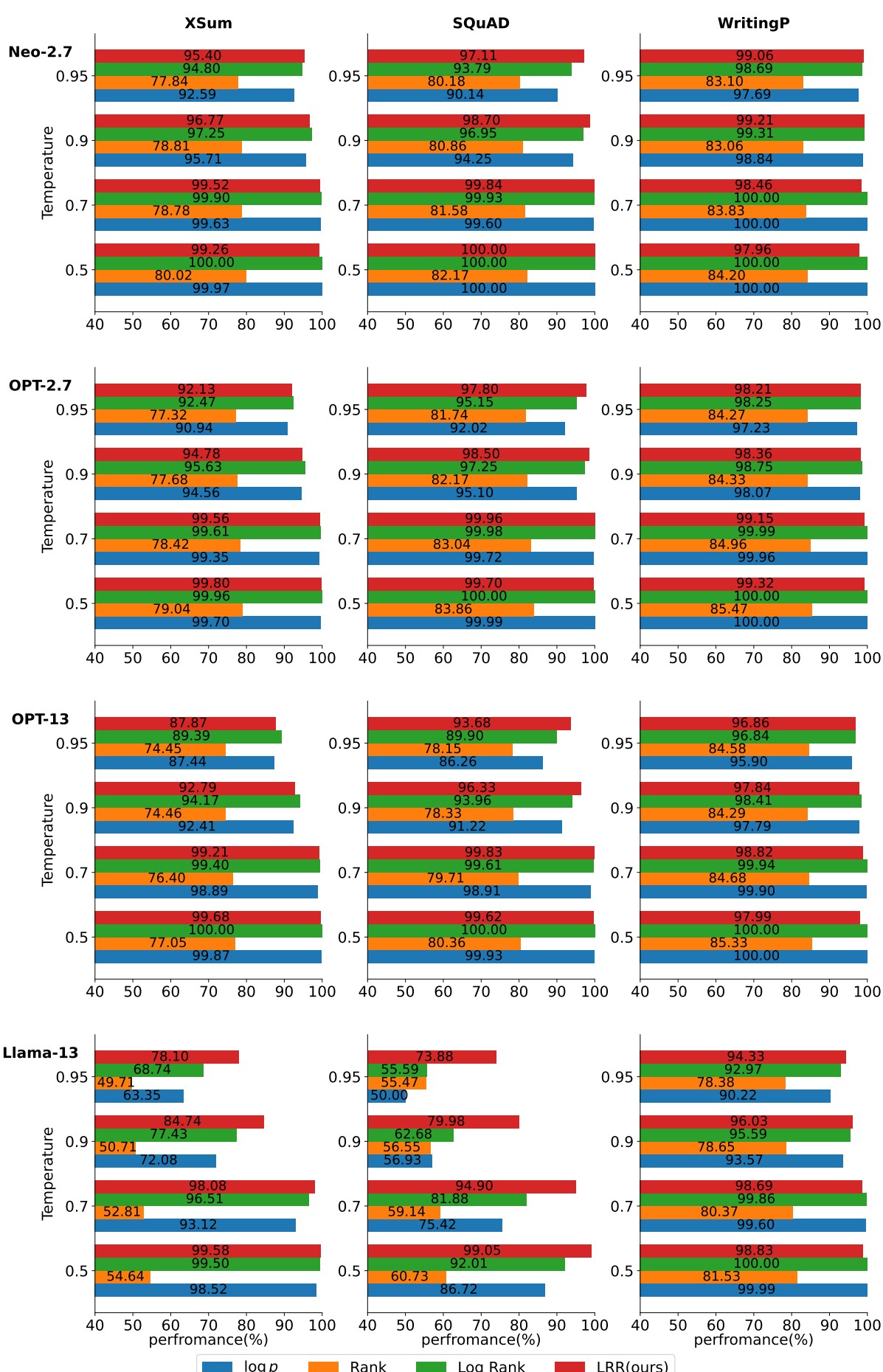

Figure 9: Comparison of perturbation-free methods using different temperatures (AUROC score).

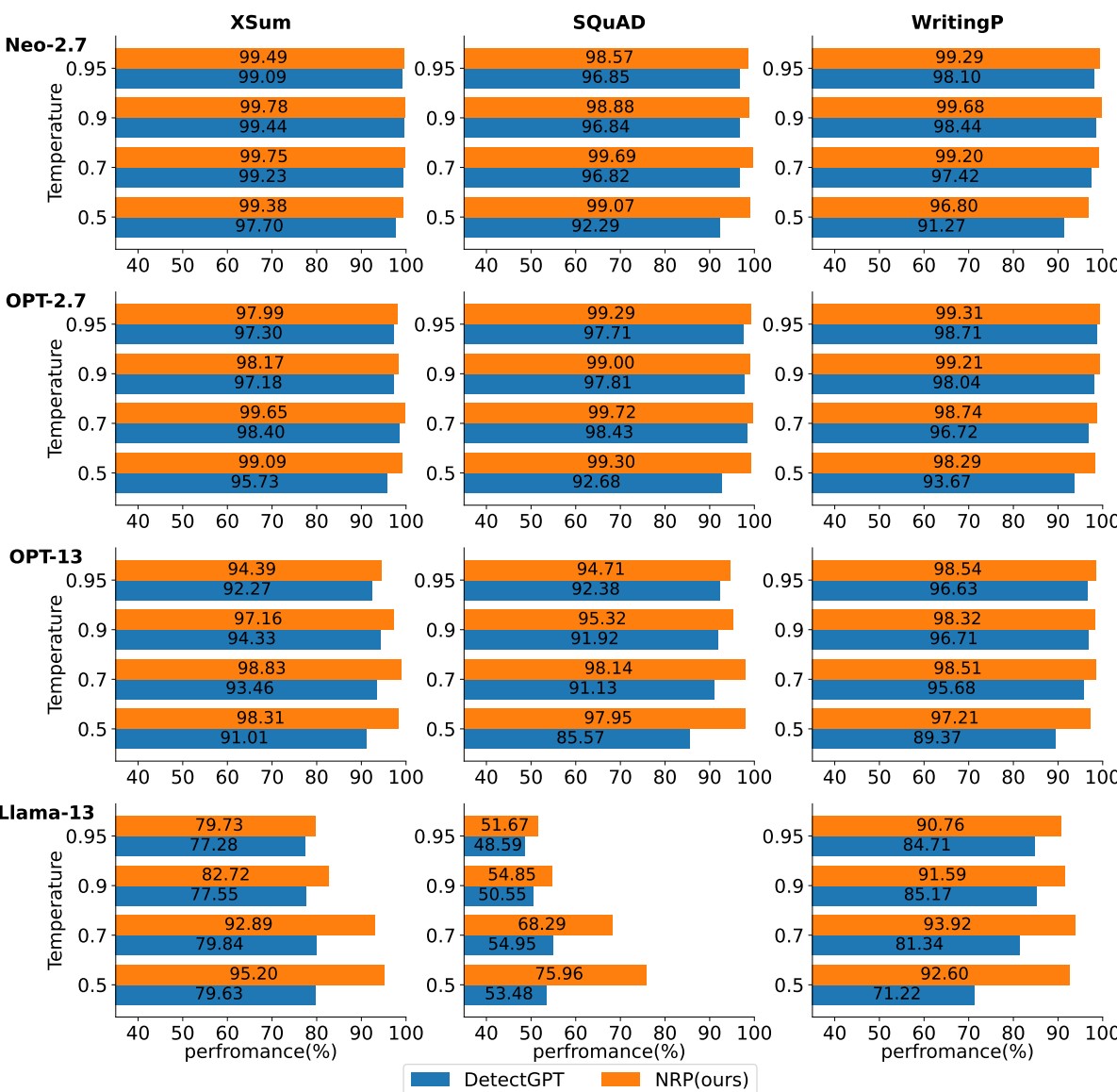

Figure 10: Comparison of perturbation methods using different temperature (AUROC score).