# OpenReview forum: "DetectLLM: Leveraging Log Rank Information for Zero-Shot Detection of Machine-Generated Text"
_EMNLP/2023/Conference — EMNLP 2023 Findings_

### Official Review · Reviewer_712P · 2023-07-31

**Soundness:** 3

**Excitement:**

3: Ambivalent: It has merits (e.g., it reports state-of-the-art results, the idea is nice), but there are key weaknesses (e.g., it describes incremental work), and it can significantly benefit from another round of revision. However, I won't object to accepting it if my co-reviewers champion it.

**Paper Topic And Main Contributions:**

This paper introduces two zero-shot methods for detecting machine-generated text using log rank information. The proposed methods can prevent malicious use such as plagiarism, misinformation, and propaganda. The paper discusses the efficiency-performance trade-off based on users' preference and provides intuition for using the methods effectively. The experiments show that the proposed methods improve over the baselines.

**Questions For The Authors:**

Can you provide more details on the limitations of your proposed methods? For example, are there certain types of machine-generated text that are more difficult to detect than others?

**Reasons To Accept:**

1.This paper introduces two novel approaches, DetectLLM-LRR and DetectLLM-NPR, for detecting machine-generated text. These methods leverage log rank information and outperform baselines in terms of accuracy and efficiency.
2. The authors provide insights on the efficiency-performance trade-off and show that their methods are more practical for real-world use.

**Reasons To Reject:**

1. There are already a number of applications for detecting AI-generated text, e.g., https://copyleaks.com/ai-content-detector, https://writer.com/ai-content-detector/. Even OPENAI released a classifier for indicating AI-written text. However, authors don't seem to make comparisons.
2.The authors use the area under the receiver operating characteristic curve (AUROC) as the sole evaluation metric for their methods. While this is a common metric for zero-shot detection, it may not capture all aspects of performance, such as precision and recall.
3. The experiments in this paper only test the proposed methods on datasets in English. It is unclear how well these methods would perform on other languages.


**Reproducibility:**

4: Could mostly reproduce the results, but there may be some variation because of sample variance or minor variations in their interpretation of the protocol or method.

**Reviewer Confidence:**

2: Willing to defend my evaluation, but it is fairly likely that I missed some details, didn't understand some central points, or can't be sure about the novelty of the work.

---

> ### Author Rebuttal · Authors · 2023-08-29
>
> Thank you for your valuable comments. Hope the following rebuttal addresses your concerns.
>
> ---
>
> ## Question 1.
>
> *There are already a number of applications for detecting AI-generated text,
> e.g., https://copyleaks.com/ai-content-detector, https://writer.com/ai-content-detector/. Even
> OPENAI released a classifier for indicating AI-written text. However, authors don’t seem to
> make comparisons.*
>
> ## Response 1.
>
> We respectfully disagree with your opinion for the following reasons.
>
> First, our methods are zero-shot, so, to make fair comparisons, we should mostly compare with other state-of-the-art zero-shot methods [3]. Note that we have compared with 5 state-of-the-art zero-shot methods in our paper.
>
> Secondly, though there exists a lot of AI-generated text detectors used commercially such as
> https://copyleaks.com/ai-content-detector, https://writer.com/ai-content-detector/ as you mentioned, but they
> are not the state-of-the-art methods for detecting machine generated texts and  **they performs poorly**. Our paper is more likely to be questioned if we compare with the methods you mentioned rather than the state-of-the-art zero-shot methods. We didn’t compare openAI classifiers for the same reason. In fact, **OpenAI’s classifier has been closed now because of their low detection accuracy**(See the article here: https://searchengineland.com/openai-ai-classifier-no-longer-available-429912).
>
> Thirdly, besides the zero-shot baselines, we have indeed provided comparison with supervised methods in the appendix of our original paper. Although this comparison is unfair to our methods, we still provided it in our paper.
>
> Note that supervised methods need training data while our methods don’t. Moreover, supervised methods tend to overfit and can’t generalize well [3]. Every time when the model updates, or changes to a new domain, supervised methods have to be retrained. Thus, our methods are more prctical since LLM updates frequently, and it is unrealistic to train a new detection model for every LLM and every domain. Meanwhile, our methods don’t attach to any specific model or domain and are more robust and realistic. As a result, we believe our research (also the whole research area of “detecting machine generated texts”) is both valuable and meaningful.
>
> ---
>
> ## Question 2.
>
> *The authors use the area under the receiver operating characteristic curve (AUROC) as the sole evaluation metric for their methods. While this is a common metric for zero-shot detection, it may not capture all aspects of performance, such as precision and recall.*
>
> ## Response 2.
>
> Firstly, it is difficult for us to illustrate multiple metrices without confusing the readers. Note that we already have multiple dense tables and graphs  (with 7 models, 7 baselines, 3 datasets, as well as a lot of ablation and parameter related experiments).
> If we provide multiple metrices, the reader might wonder which metric should they look at and got lost in all these tables and metrics).
>
> Moreover, since detection rates are heavily dependent on the chosen detection threshold, the AUC-ROC metric
> is more commonly used to measure detector performance , **which considers the range of all possible thresholds**. See [1, 2, 3]. Meanwhile, recall and precision are rarely used in previous work of detecting machine generated texts. We have to set thresholds ourselves if when calculate recall or precision, and the result would heavily depend on the thresholds we set. (Imagine that you can easily manipulate the experimental results by setting a "not so good" thresholds...). Thus, precision and recall is not a good evaluation metric in our setting.
>
>
>
>
>
> However, since you have mentioned, we provide experiments on GPT2-xl generated texts using TPR under 0.01 FPR [3] for your reference.
> TPR under certain FPR might also be an important metrics [3], we will add the above results in the revised version of our paper to address your concern.
>
> The results of TPR with 0.01 FPR are showing below:
>
> For perturbation based methods: DetectGPT and NPR:
>
> |   Dataset          | SquAD|   Xsum |WritingP  |
> |:----------:|:----------------------:|:----------------------:|:----------------------:|
> | DetectGPT |                 98.45 |                 98.73 |                 99.11 |
> | NPR(ours)       |                 **99.31** |                 **99.48** |                 **99.76** |
>
>
> For non-perturbation based methods:
> |   Dataset          | SquAD|   Xsum |WritingP  |
> |:----------:|:----------------------:|:----------------------:|:----------------------:|
> | rank      |                 83.46 |                 79.79 |                 87.62 |
> | $\log p$  |                 90.72 |                 89.16 |                 96.71 |
> | Log rank  |                 94.33 |                 91.75 |                 98.02 |
> | entropy   |                 57.97 |                 56.78 |                 36.45 |
> | LRR(ours)       |                 **97.42** |                 **93.47** |                 **98.34** |
>
> It can be seen from the above result that the TPR  under FPT fixed as 0.01 is consistent with our previous used AUC metric. Our methods still achieve the state of the art performance among the baseline. Moreover, we will add this metric in our codes for reproducibility.
>
> ---
>
> ## Question 3.
> *The experiments in this paper only test the proposed methods on datasets in English. It is unclear how well these methods would perform on other languages.*
>
>
> ## Response 3.
>
>
> We only use English texts because most of the LLMs are not multi-lingual. For example, the 7 LLMs we experimented with can only generate English texts. There aren't many multilingual LLMs that are large enough to generate indistinguishable texts in other languages.
>
> However, to address the reviewer’s concern, we add additional experiments with German texts from WMT16 dataset as human written texts, and apply m-GPT to generating the machine generated counterparts. For baseline methods DetectGPT and NPR, which requires multi-lingual perturbation model, we apply mT5-xl for perturbation. The rest of the settings are the same as for English texts.
>
> The result on German can be seen in the Tables below.
>
> For perturbation based methods(DetectGPT and NPR):
>
> | name      |   AUC |
> |:----------:|:------:|
> | DetectGPT | 96.39 |
> | NPR(ours)       | **98.46** |
>
> For perturbation-free methods:
>
> | name      |   AUC |
> |:----------:|:------:|
> | rank      | 83.81 |
> | $\log p$  | 81.67 |
> | Log rank  | 87.47 |
> | entropy   | 58.61 |
> | LRR(ours)       | **95.59** |
>
> It can been seen from above table that our methods still hold advantages over other baselines. Actually, this result is intuitive because the statistics used in our methods are agnostic to the language or the domain. As long as the LLM has the ability to generate text in a specific language, the statistics given by the LLM would reveal whether the text is generated by the model or not. We will add this experiments in camera ready version and release the code for this muli-lingual experiments as well.
>
> ---
>
>
> ## Question 4.
>
> *Can you provide more details on the limitations of your proposed methods? For example, are there certain types of machine-generated text that are more difficult to detect than others*
>
> ## Response 4.
>
> We are not sure if there are certain types of machine generated texts that are more difficult to detect. with our methods. Unlike supervised methods, which don't work well on out-of-domain data,  our methods don’t require training so it generally works well for all types of data.
>
> However, according to our experiments, we found that most of the detection methods (both the baselines and our methods) have a slightly performance drop on Llama-13b generated data, which is probably because Llama-13b is more aligned with human generated texts and thus is harder to detect. As a result, we think that the performance depends on the LLMs, while other methods might depend on both the LLMs and the types of data.
>
> ---
>
> # References
>
> [1] Xinlei He, Xinyue Shen, Zeyuan Chen, Michael Backes, and Yang Zhang. Mgtbench:
> Benchmarking machine-generated text detection. arXiv preprint arXiv:2303.14822, 2023.
>
> [2] Kalpesh Krishna, Yixiao Song, Marzena Karpinska, John Wieting, and Mohit Iyyer. Para-
> phrasing evades detectors of ai-generated text, but retrieval is an effective defense. arXiv
> preprint arXiv:2303.13408, 2023.
>
> [3] Eric Mitchell, Yoonho Lee, Alexander Khazatsky, Christopher D Manning, and Chelsea
> Finn. Detectgpt: Zero-shot machine-generated text detection using probability curvature.
> arXiv preprint arXiv:2301.11305, 2023.

---

### Official Review · Reviewer_CKfz · 2023-08-03

**Soundness:** 3

**Excitement:**

3: Ambivalent: It has merits (e.g., it reports state-of-the-art results, the idea is nice), but there are key weaknesses (e.g., it describes incremental work), and it can significantly benefit from another round of revision. However, I won't object to accepting it if my co-reviewers champion it.

**Paper Topic And Main Contributions:**

This paper proposes two novel methods for detecting text generated from LLMs through either Log-Likelihood Log-Rank ratio (LRR) or Normalized perturbed log-rank (NPR). The experiments show their effectiveness over previous baselines on various datasets and models. Since their zero-shot method is very competitive with previous training-based methods, this paper would offer a good solution to the emerging issue of misinformation spread on the internet.

**Questions For The Authors:**

q1: Table 1: all results on Llama 13B seem much worse than on other models. Do you have any idea why this happens?

q2: Line 201: it is unclear what rank means here at first glance. Perhaps give a formal definition?

q3: Line 223: I don't know what r(x) means compared with r(xi|x<i). Does it mean you add all tokens’ ranks in a sequence x?

q4: in the original DetectGPT, they tested the black-box setting with a surrogate model and found it to be somewhat useful. I am curious whether your methods can also achieve this since we do not have access to many closed models.

q5: except for the AUROC score, what is the TPR under certain FPR(like 1%) since this is more close to realistic use?


**Reasons To Accept:**

the proposed methods are simple but useful, outperforming various baselines. The experimental settings are reasonable.

They test various models and datasets to validate their methods under different decoding schemes such as top-p, top-k, or temperature.

the overall writing and demonstrations are clear and easy to follow.


**Reasons To Reject:**

There is a lack of experiments to show the robustness of their methods, for example, how would their method perform when the text is rephrased by either human or machine?

It is unclear whether their method also works for other languages except English since they only test it in English text.

See more in questions.

**Reproducibility:**

4: Could mostly reproduce the results, but there may be some variation because of sample variance or minor variations in their interpretation of the protocol or method.

**Reviewer Confidence:**

5: Positive that my evaluation is correct. I read the paper very carefully and I am very familiar with related work.

**Typos Grammar Style And Presentation Improvements:**

Line 846: wrong citation?

---

> ### Author Rebuttal · Authors · 2023-08-29
>
> Thank you very much for bringing up those insightful questions. Hope the following rebuttal adresses your concerns.
>
> ---
>
> ## Question1.
>
> *How would their method perform when the text is rephrased by either human or machine?*
>
> ## Response 1.
> We do have thought of this problem in the early stage of our paper when we were designing experiments. In the end, we didn’t include these experiments because we find it controversial: when a machine generated text is paraphrased by humans or machine, we are not sure if we should still call it “machine generated text”, meanwhile, when human-written text is rephrased by machine, we are not sure if we can still label it as“human-written”. Thus, to keep our paper structurally clear, we only consider classify two formally defined classes: machine generated text, and human written one.
>
> Put the above controversy aside, if we rephased machine/human written text while still considering them as machine-generated/ human written, intuitively, the performance will drop [4] for all detection methods, not just ours. Note that this performance drop is reasonable and is not the weakness of our method, because these rephased texts don't satisfy our definition of machine generated/human written texts used in our paper. Our definition of machine/human written texts assumes that texts are directly generated from machine or human.
>
> Although the robustness is not a main topic of this work, we would like to share some preliminary thoughts on how to reframe this problem and propose new methods on this setting by integrating our methods for the future work.
>
> Firstly, we reframe this problem as a 3 class classification problem.
>
> (i). Machine generated text without rephase
>
> (ii). machine generated text paraphrased by a third model
>
> (iii). Human written texts
>
> The above three classes is more clearly defined to avoid unnecessary confusion. We didn’t add “human written texts paraphrased by a third model” due to practical considerations: people are more likely to paraphrase machine generated texts to avoid being detected by detectors.
>
> For class (i), the feature is quite distinguishable and it should be easy to categorize class (i) correctly with features used in our paper. For class (ii) and (iii), they might both have low LLR and NPR. To distinguish (ii) and (iii), we can design methods to identify whether a text has been paraphrased or not, which corresponds to another line of research called paraphrase detection [1, 8].
>
>
> ---
>
> ## Question 2.
>
> *Unclear whether their method also works for other languages except English since they only test it in English text.*
>
> ## Response 2.
>
> We only use English texts because most of the LLMs are not multi-lingual. For example, the 7 LLMs we experimented with can only generate English texts. There aren't many multilingual LLMs that are large enough to generate indistinguishable texts in other languages.
>
> However, to address the reviewer’s concern, we add additional experiments with German texts from WMT16 dataset as human written texts, and apply m-GPT to generating the machine generated counterparts. For baseline methods DetectGPT and NPR, which requires multi-lingual perturbation model, we apply mT5-xl for perturbation. The rest of the settings are the same as for English texts.
>
> The result on German can be seen in the Tables below.
>
> For perturbation based methods(DetectGPT and NPR):
>
> | name      |   AUC |
> |:----------:|:------:|
> | DetectGPT | 96.39 |
> | NPR(ours)       | **98.46** |
>
> For perturbation-free methods:
>
> | name      |   AUC |
> |:----------:|:------:|
> | rank      | 83.81 |
> | $\log p$  | 81.67 |
> | Log rank  | 87.47 |
> | entropy   | 58.61 |
> | LRR(ours)       | **95.59** |
>
> It can been seen from above table that our methods still hold advantages over other baselines. Actually, this result is intuitive because the statistics used in our methods are agnostic to the language or the domain. As long as the LLM has the ability to generate text in a specific language, the statistics given by the LLM would reveal whether the text is generated by the model or not. We will add this experiments in camera ready version and release the code for this muli-lingual experiments as well.
>
> ---
>
> ## Question 3.
>
> *Table 1: all results on Llama 13B seem much worse than on other models. Do you have any idea why this happens?*
>
> ## Response 3.
>
> As shown in the original LLama paper [7], the LLama-13B generated texts more align with human-written texts and experimentally outperforms OPT and GPT-3 175B on most benchmarks, thus, it is conceivable that machine generated text detectors achieve low detection rate on LLama-13b.
>
> ---
>
> ## Question 4.
>
> *Line 201: it is unclear what rank means here at first glance. Perhaps give a formal definition?*
>
> ## Response 4.
>
> Thanks for mentioning this, we will provide more explanation in the revised version.
> “rank” means the rank of the next token probability given previous tokens. For example, if one token has the highest probability, its rank is 1, and the rank of token with the second highest probability has rank 2, and so on [2].
>
> Formally, given the previous tokens $x_1, \cdots, x_{i-1}$, and given the tokenization of the vocabularies $v_1, v_2,\cdots, v_m$, we can calculate the probability of all the tokenized vocabularies conditioned on previous token $p(v_1|x_{<i}),  p(v_2|x_{<i}), \cdots, p(v_m|x_{<i})$, the rank of $x_i$ is defined as the rank of $p(x_i|x_{<i})$ in  $p(v_1|x_{<i}),  p(v_2|x_{<i}), \cdots, p(v_m|x_{<i})$ after sorting them in descending order.
>
> ---
>
>
> ## Question 5.
>
> *Line 223: I don’t know what r(x) means compared with r(xi—x¡i). Does it mean you add all tokens’ ranks in a sequence x?*
>
> ## Response 5.
>
> Thanks for bringing this up, you are correct!
>
> Note that $x$ is a text while $x_i$ is the $i$-th token, thus, $r(x)$ represents the rank of the text $x$, which is the average rank of all tokens in $x$.
>
> ---
>
>
> ## Question 6.
>
> *in the original DetectGPT, they tested the black-box setting with a surrogate model and found it to be somewhat useful. I am curious whether your methods can also achieve this since we do not have access to many closed models.*
>
> ## Response 6.
>
> Thank you for bringing this up. In fact, we have mentioned this in our “Limitation and future work” section!
> The limitation of our zero-shot methods can be addressed by using ensemble methods such as weak supervision [6] to aggregate statistics from other open-source LLMs rather than using the single source model. In this case, it would be totally black-box, and we don’t have to know which source model the text is from and we don’t need access to the source model statistics.
>
> Since the main purpose of our method is to bring up novel statistics for machine generated text detection, in our paper, we didn’t do additional experiments on that (otherwise, we might confuse the readers about the main contribution of our work).
>
> However, we would like to elaborate more on the feasibility of the above idea. The simplest example is majority vote, if we have several detectors (or, use the term in weak supervision: labelers), and each detector has a detection accuracy higher than random guess, when we aggregate these labelers, the final classifier can be boosted. (See more weak supervision algorithms in the github repo of the paper [9]). Weak supervision can convert a weak/inaccurate classifier to a strong/accurate classifier as long as the original classifiers are better than random. Note that using other models, the accuracy will decrease but the detection rate is still better than random and thus we could use statistics from surrogate models and aggregate them to boost the detection accuracy.
>
> However, the main concern we have here is not the accuracy but efficiency. For the proposed LLR, this aggregation should be fine, but for perturbation based methods such as DetectGPT [5] and NPR, the user might have to balance the efficiency with accuracy, since perturbation based method itself already takes much time and computational resources, using weak supervised methods would increase the time at least linearly with the number of surrogate models. For example, if you use 5 several weak labels, it would be at least 5 times more computationally expensive to get an accuracy boosted final classifier.
>
> Using only one surrogate model is also okay, but it can't achieve that high accurancy compared to using the source model. (However, it could possibly achieve a comparable accurancy when using multiple surrogate models and aggregate them as we discussed above.)
>
>
> ---
>
> ## Question 7.
>
> *Except for the AUROC score, what is the TPR under certain FPR(like 1%)*
>
> ## Response 7.
>
> Due to time limits, we provide experiments on GPT2-xl generated texts and the following Tables shows TPR under 0.01 FPR.
>
> For perturbation based methods: DetectGPT and NPR:
>
> |   Dataset          | SquAD|   Xsum |WritingP  |
> |:----------:|:----------------------:|:----------------------:|:----------------------:|
> | DetectGPT |                 98.45 |                 98.73 |                 99.11 |
> | NPR(ours)       |                 **99.31** |                 **99.48** |                 **99.76** |
>
>
> For non-perturbation based methods:
> |   Dataset          | SquAD|   Xsum |WritingP  |
> |:----------:|:----------------------:|:----------------------:|:----------------------:|
> | rank      |                 83.46 |                 79.79 |                 87.62 |
> | $\log p$  |                 90.72 |                 89.16 |                 96.71 |
> | Log rank  |                 94.33 |                 91.75 |                 98.02 |
> | entropy   |                 57.97 |                 56.78 |                 36.45 |
> | LRR(ours)       |                 **97.42** |                 **93.47** |                 **98.34** |
>
>
> We didn't contain it in our paper because AUC is a more commonly used metric for machine generated text detection [3, 4, 5] and we don't want to confuse readers with multiple metrics since our tables and graphs are already too dense. Meanwhile, as you mentioned in the review, TPR under certain FPR is also an important metrics [4], we will add the above results in the revised version of our paper.
>
> ---
>
> ## Question 8.
> *Line 846: wrong citation?*
>
> ## Response 8.
>
> Thanks for pointing it out, the more accurate citation should be this two papers: [2, 5]. However, our original citation aligns with the citation used in paper [3].
>
> ---
>
> # References
>
> [1] Jonas Becker, Jan Philip Wahle, Terry Ruas, and Bela Gipp. Paraphrase detection: Human vs. machine content. arXiv preprint arXiv:2303.13989, 2023.
>
> [2] Sebastian Gehrmann, Hendrik Strobelt, and Alexander M Rush. Gltr: Statistical detection and visualization of generated text. arXiv preprint arXiv:1906.04043, 2019.
>
> [3] Xinlei He, Xinyue Shen, Zeyuan Chen, Michael Backes, and Yang Zhang. Mgtbench: Benchmarking machine-generated text detection. arXiv preprint arXiv:2303.14822, 2023.
>
> [4] Kalpesh Krishna, Yixiao Song, Marzena Karpinska, John Wieting, and Mohit Iyyer. Para- phrasing evades detectors of ai-generated text, but retrieval is an effective defense. arXiv preprint arXiv:2303.13408, 2023.
>
> [5] Eric Mitchell, Yoonho Lee, Alexander Khazatsky, Christopher D Manning, and Chelsea Finn. Detectgpt: Zero-shot machine-generated text detection using probability curvature. arXiv preprint arXiv:2301.11305, 2023.
>
> [6] Alexander J Ratner, Christopher M De Sa, Sen Wu, Daniel Selsam, and Christopher R ́e. Data programming: Creating large training sets, quickly. Advances in neural information processing systems, 29, 2016.
>
> [7] Hugo Touvron, Thibaut Lavril, Gautier Izacard, Xavier Martinet, Marie-Anne Lachaux, Timoth ́ee Lacroix, Baptiste Rozi`ere, Naman Goyal, Eric Hambro, Faisal Azhar, et al. Llama: Open and efficient foundation language models. arXiv preprint arXiv:2302.13971, 2023.
>
> [8] Jan Philip Wahle, Terry Ruas, Tom ́aˇs Folt`ynek, Norman Meuschke, and Bela Gipp. Identi- fying machine-paraphrased plagiarism. In International Conference on Information, pages 393–413. Springer, 2022.
>
> [9] Jieyu Zhang, Yue Yu, Yinghao Li, Yujing Wang, Yaming Yang, Mao Yang, and Alexan- der Ratner. Wrench: A comprehensive benchmark for weak supervision. arXiv preprint arXiv:2109.11377, 2021.

---

### Official Review · Reviewer_1rWH · 2023-08-06

**Soundness:** 4

**Excitement:**

4: Strong: This paper deepens the understanding of some phenomenon or lowers the barriers to an existing research direction.

**Paper Topic And Main Contributions:**

This paper introduces two novel zero-shot methods, namely DetectLLM-LRR and DetectLLM-NPR, for detecting machine-generated text. These methods leverage log rank information and exhibit distinct characteristics in terms of efficiency and accuracy.

**Reasons To Accept:**

1. Comprehensive experimental evaluations on three datasets and seven language models provide a thorough analysis of the proposed methods' performance. The results demonstrate a remarkable improvement over the state-of-the-art by 3.9 and 1.75 AUROC points absolute, highlighting the efficacy and superiority of the proposed techniques.
2. The authors also showcase the practicality of DetectLLM-NPR, which requires fewer perturbations than previous work while achieving similar performance levels. This feature makes it more suitable for real-world use, where efficiency is essential in detecting machine-generated text in social media and education settings. The consideration of efficiency-performance trade-offs and providing insights for effective practical usage further strengthen the paper's quality and relevance to real-world applications.

**Reasons To Reject:**

As mentioned in the paper, one limitation of the zero-shot methods is that it requires statistics from the source model, which doesn't apply to the cases when one has no access to source models. However, this is not a weakness of the proposed methods.

**Reproducibility:**

4: Could mostly reproduce the results, but there may be some variation because of sample variance or minor variations in their interpretation of the protocol or method.

**Reviewer Confidence:**

1: Not my area, or paper was hard for me to understand. My evaluation is just an educated guess.

---

> ### Author Rebuttal · Authors · 2023-08-28
>
> Thank you for your thoughtful evaluation and insightful feedback! Here, we would like to further address some of your concerns mentioned in the review:
>
> ---
>
> ## Question 1.
>  *Zero-shot methods require statistics from the source model.*
>
> ## Response 1.
>
> We thank the reviewer for acknowledging that this limitation applies to all the zero-shot methods, rather than being specific to our methods. We will add this point to "Limitation" section.
>
> Indeed, as mentioned in our "Limitation and future work" section of our paper, this limitation can be addressed with ensemble methods such as weak supervision [1] to aggregate statistics from other open-source LLMs rather than using the single source model. In this case, it would be totally black-box, and we don’t have to know which source model the texts are from and we don’t need access to the source model statistics.
>
> Furthermore, there is a surge in developing open LLMs [2] over closed LLMs. Considering the large adoption of these open LLMs, our work can be effective in detecting the open-LLM-generated text.
>
> Lastly, since LLM model producers should be responsible for their model, even for close-sourced models, the LLM model producers can integrate our method for detection. Thus, our method is still valuable to the detecting machine generated text research area.
>
> ---
>
> # References:
> [1] Alexander J Ratner, Christopher M De Sa, Sen Wu, Daniel Selsam, and Christopher R ́e. Data
> programming: Creating large training sets, quickly. Advances in neural information processing
> systems, 29, 2016.
>
> [2] Wayne Xin Zhao, Kun Zhou, Junyi Li, Tianyi Tang, Xiaolei Wang, Yupeng Hou, Yingqian Min,
> Beichen Zhang, Junjie Zhang, Zican Dong, et al. A survey of large language models. arXiv
> preprint arXiv:2303.18223, 2023

---

### Meta-Review · Area_Chair_oRCX · 2023-09-13

**Recommendation:** 3

**Metareview:**

**Strengths**:

1. Critical problem of detecting whether text is machine generated or human generated.

2. Two novel approaches, DetectLLM-LRR and DetectLLM-NPR. Comprehensive experimental evaluations on three datasets and seven language models. Expts with different decoding schemes such as top-p, top-k, and temperature.

3. The proposed DetectLLM-NPR requires fewer perturbations than previous work while achieving similar performance levels.


**Weaknesses**:

1. Anonymous code not shared.

2. It would be interesting to look at certain types of machine-generated text (e.g., text in healthcare or law) that could be more difficult to detect than others.

**Suggestions**:

1. Thanks for putting up results for German. However, since the majority of the evaluation is for English only, limitations should clearly state that the work is for English only.

2. It would be interesting to look at certain types of machine-generated text (e.g., text in healthcare or law) that could be more difficult to detect than others.

3. Please incorporate rebuttal content into the main text.

---

### Decision · Program_Chairs · 2023-10-07

**Decision:**

Accept-Findings

**Comment:**

**Strengths**:

1. Critical problem of detecting whether text is machine generated or human generated.

2. Two novel approaches, DetectLLM-LRR and DetectLLM-NPR. Comprehensive experimental evaluations on three datasets and seven language models. Expts with different decoding schemes such as top-p, top-k, and temperature.

3. The proposed DetectLLM-NPR requires fewer perturbations than previous work while achieving similar performance levels.


**Weaknesses**:

1. Anonymous code not shared.

2. It would be interesting to look at certain types of machine-generated text (e.g., text in healthcare or law) that could be more difficult to detect than others.

**Suggestions**:

1. Thanks for putting up results for German. However, since the majority of the evaluation is for English only, limitations should clearly state that the work is for English only.

2. It would be interesting to look at certain types of machine-generated text (e.g., text in healthcare or law) that could be more difficult to detect than others.

3. Please incorporate rebuttal content into the main text.